# Exploring the relationship between telehealth utilization and treatment burden among patients with chronic conditions: A cross-sectional study in Ontario, Canada

**Farah Tahsin**[1]*, **Carolyn Steele Gray**[1,2], **Jay Shaw**[1,3], **Aviv Shachak**[1]

**1** Institute of Health Policy, Management, and Evaluation, University of Toronto, Toronto, Ontario, Canada, **2** Lunenfeld-Tanenbaum Research Institute, Sinai Health, Toronto, Ontario Canada, **3** Department of Physical Therapy, University of Toronto, Toronto, Ontario, Canada

* farah.tahsin@mail.utoronto.ca

**Data Availability Statement:** Data leading to this study's findings is available without restriction within the manuscript.

## Abstract

One in five Canadians lives with one or more chronic conditions. Patients with chronic conditions often experience a high treatment burden because of the work associated with managing care. Telehealth is considered a useful solution to reduce the treatment burden among patients with chronic conditions. However, telehealth can also increase the treatment burden by offloading responsibilities on patients. This cross-sectional study conducted in Ontario, Canada examines the association between telehealth utilization and treatment burden among patients with chronic conditions. This study aimed to explore whether and to what extent, telehealth use is associated with treatment burden among patients with chronic conditions. The secondary objective was to explore which sociodemographic variables are associated with patients' treatment burden. An online survey was administered to community-dwelling patients with one or more chronic conditions. The Treatment Burden Questionnaire (TBQ-15) was used to measure the patient's level of treatment burden, and a modified telehealth usage scale was developed and used to measure the frequency of telehealth use. Data was analyzed using descriptive statistics, correlations, analyses of variance, and hierarchical linear regression analysis. A total of 75 patients completed the survey. The participants' mean age was 64 (SD = 18.93) and 79% were female. The average reported treatment burden was 72.15 out of 150 (a higher score indicating a higher level of burden). When adjusted for demographic variables, a higher frequency of telehealth use was associated with experiencing a higher treatment burden, but the association was not statistically significant. Additionally, when adjusted for demographic variables, younger age, and the presence of an unpaid caregiver were positively related to a high treatment burden score. This finding demonstrates that some patient populations are more at risk of experiencing high treatment burden in the context of telehealth use; and hence, may require extra support to utilize telehealth technologies. The study highlights the need for further research to explore how to minimize the treatment burden among individuals with higher healthcare needs.

**Funding:** The author(s) received no specific funding for this work.

**Competing interests:** The authors have declared that no competing interests exist.

## Author summary

Telehealth is considered as a promising solution to support patients with chronic conditions. However, using and adopting telehealth technologies can also increase/alleviate patients' workload. This workload associated with healthcare use is called treatment burden. Currently, little is known about the relationship between telehealth use and treatment burden. Therefore, we conducted a study on how telehealth can make treatment burdens better or worse for these patients. Through our survey, we found that there is a positive association between high frequency of telehealth use and treatment burden. Our study shows that younger patients and patients without unpaid caregiver experience higher level of burden; and therefore, as a health system, we need to explore how to minimize treatment burden among this patient population through leveraging technologies and social services.

## Background

According to Statistics Canada, over 14.6 million Canadians (45.1%) reported living with one or more chronic conditions in 2021 [1]. Effective management of chronic conditions is essential for maintaining optimal health and reducing further complications [2,3]. Common elements of a chronic conditions-related self-management routine include: prescribed medications, monitoring symptoms, engaging in regular exercise, and maintaining a healthy diet [4,5]. However, these activities demand significant time, energy, and effort from both patients and their support networks [6–9]. For example, one cross-sectional study found that the average time required for an adult diagnosed with type 2 diabetes to complete all recommended aspects of self-care (including exercise) was approximately 4 hours per day [6]. Overall, there is a growing recognition of this healthcare-imposed workload on patients and their support networks. This workload is referred to as the *treatment burden* [10].

Patients with chronic conditions face various types of treatment burdens which include medication, administrative, lifestyle, financial, and relational burdens [11,12]. The magnitude and type of burden vary based on social circumstances, capacity, and healthcare-related factors [13]. More specifically for this study, socioeconomic status (SES) plays an important role in the type and level of treatment burden experienced by patients and their caregivers [14,15]. SES is a resource-based measure, which defines a person's social position [16]. It is often measured by a person's income, education, or occupational status [16]. SES plays a significant role in an individual's access to health and socioeconomic resources, and consequently to their health outcomes [16,17]. Patients with low SES often experience a heightened level of burden, specifically financial burden [18]. In the context of this study, the association between low SES and treatment burden could be attributed to several factors including high symptom burden [19], low digital or health literacy [18], and system-level barriers (e.g. poor access to coordinated care, poor access to social care, digital divide) [20].

Telehealth-based technologies are commonly used to mitigate treatment burden-related challenges [21]. Telehealth is a broad term. For this study, we have adopted the New England Journal of Medicine's definition of telehealth, which is: *the use of technology to provide remote personalized health care to patients.* [22] This broad definition of telehealth includes virtual care, patient portals, telemonitoring devices as well as self-management apps. These technologies enable remote care, facilitating disease monitoring without the constraints of frequent

office visits [21]. Despite these benefits, it is essential to acknowledge that integrating telehealth can sometimes exacerbate the treatment burden [23].

For example, poorly integrated health technologies, challenging interfaces, low information quality, or a high digital literacy requirement can worsen the burden for patients [24]. Clinicians are increasingly concerned about the potential treatment burden telehealth may pose, especially for patients with low socioeconomic status (SES) [25,26]. This population is more likely to have low digital literacy, which could make telehealth use particularly burdensome for them [27].

While there has been some recognition of disparity in experienced treatment burden among patients with chronic conditions in the context of telehealth use [14,15], there is a lack of knowledge of the relationship between telehealth use and treatment burden among patients with chronic conditions [28]. Additionally, there is a considerable lack of research on the specific types of treatment burden faced by this patient population and effective strategies for its mitigation. Moreover, there is a lack of research distinguishing between necessary burdens, such as exercise that positively contributes to patients' health, and avoidable burdens, such as administrative tasks associated with paperwork. Telehealth can help mitigate avoidable burdens by offering technologies such as online patient portals, which reduce the need for patients to complete paperwork for providers. Additionally, telehealth can support health-promoting activities by providing virtual self-management programs and classes. However, there is insufficient evidence on the overall impact of telehealth on treatment burden [28].

The objective of this study was to explore the relationship between telehealth use and treatment burden among patients with chronic conditions. The secondary objective of this paper was to explore the demographic characteristics associated with high treatment burden.

## Research questions

To explore the relationship between telehealth use and treatment burden, this paper addressed the research question: *to what extent is the use of telehealth associated with treatment burden among patients living with chronic conditions*?

Given the potential role of SES in the experience of treatment burden, an additional sub-question is added which asks: *Which sociodemographic factors are associated with patients' treatment burden scores*?

## Method

To answer the research questions, we conducted a cross-sectional survey in Ontario, Canada. The ethical approval for this study was received from the University of Toronto Research Ethics Board. All participants provided their written consent before completing the survey.

**Fig 1** shows the conceptual framework for this cross-sectional study. The confounder variables for this study were selected through a literature review to ensure the collection of known demographic factors that are correlated with either outcome or independent variables.

Based on the previous literature, this study had three preliminary hypotheses:

**H1:** Patients who have used telehealth for their healthcare needs will have a lower treatment burden since previous studies show that telehealth may reduce their time, cost, and travel burden [29]

H1a: More specifically, telehealth use will be positively associated with treatment burden that is beneficial for health (i.e., following medication regimen and lifestyle) but negatively associated with administrative and travel burden.

Covariates: Age, sex/gender, ethnicity, income level, household income, education, occupation status, number of chronic conditions, types of chronic conditions

The frequency of telehealth use in last 12 months

Outcome variable: Treatment Burden Score

**Fig 1. Conceptual framework for the study.**

**H2:** Both telehealth use and treatment burden will be influenced by patient characteristics such as the number of co-morbidities, and SES.[14,29–31]

## Study population and setting

The eligibility criteria for the study participation included: (1) Patients age 18 or over who have one or more chronic condition(s) regardless of whether they used telehealth or not, (2) living in the community. Participants were excluded if they were: (a) residents in care homes, (b) individuals receiving palliative care, (c) diagnosed with severe mental health conditions impacting consent or questionnaire response.

Participants were recruited from Community Health Centers (CHCs) across Ontario and also through social media pages. CHCs are specialized primary care models, which primarily serve populations in lower-income neighbourhoods and specific structurally marginalized populations including individuals living in poverty, immigrant communities, and sex workers [32]. Given this study's focus on structurally marginalized populations, we intentionally recruited patients from CHCs.

**Participant recruitment procedure through CHCs.** The Alliance for Healthier Communities (AOHC) [32], a nonprofit primary care network that represents CHCs in Ontario, initiated the study recruitment by sending invitation letters to executive directors of CHCs throughout the province. Interested CHC leaders contacted the principal investigator, FT, who then met with each CHC team which included clinic managers and one or two front-line staff. In these meetings, FT presented study details and eligibility criteria for patients. Following this, primary care managers conveyed the study information to other front-line staff, including nurse practitioners and allied health professionals leading patient education or self-management groups (e.g., diabetes education group, COPD support groups) in each CHC. These professionals distributed information packages, including a study letter and the principal investigator's contact details to eligible patients and caregivers identified for the study. Study posters and brochures were also displayed in the lobby and reception areas of each participating CHC. Interested participants contacted the principal investigator and underwent a pre-screening interview by phone.

**Social media recruitment strategies.** To reach an adequate sample size, additional participants were recruited through FT's social media (Twitter and LinkedIn) as well as targeted social media pages such as the Young Caregiver Association [33], and the Ontario Caregiver Organization [34]. Given the recruitment challenges associated with structurally marginalized communities [35], this sort of social media-based recruitment strategy is becoming widely used in community-based research studies.

## Survey distribution procedure

Once participants contacted the first author (FT) to participate in the study, after confirming eligibility, FT provided them with the consent form and an online survey link to fill out the survey. The survey links were generated and administered through Redcap software—a secure web application for online surveys. Patients' survey responses were collected and stored in Redcap ©. Although a paper copy of the survey was offered in the recruitment materials, all participants preferred an online survey option.

## Measurement instruments

The survey package included the measures of treatment burden, frequency of telehealth uses in the last 12 months, and sociodemographic information.

**Treatment burden.** We assessed patients' treatment burden using the Treatment Burden Questionnaire (TBQ; S1 Appendix) [36].

The TBQ was developed to assess the treatment burden of patients who live with one or more chronic conditions [36]. The five dimensions of TBQ-15 and associated questions with each one are described below in **Box 1**.

---

Box 1. The five dimensions of TBQ-15 and associated questions within each dimension [36]

Dimension 1: Medications burden

a) The problems related to the taste, shape or size of your tablets and/or the annoyance caused by your medications

b) Number of times you should take your medication daily

c) The efforts you make not to forget to take your medications

d) The necessary precautions when taking your medications

Dimension 2: Administrative Burden

a) The problems related to lab tests and other exams (frequency, time spent and associated with nuisances or inconveniences

b) The problems related to self-monitoring frequency, time spent and associated nuisances or inconveniences.

c) The problem related to doctor visits and other appointments: frequency and time spent for these visits and difficulties finding healthcare providers

d) How would you rate the problems related to the difficulties you could have in your relationships with healthcare providers (for example: feeling not listened to enough or not taken seriously)

---

e) The problems related to arranging medical appointments (doctors' visits, lab tests and other exams) and reorganizing your schedule around these appointments

f) The problems related to the administrative burden related to healthcare (for example: all you have to do for hospitalizations, reimbursements and/or obtaining social services)

Dimension 3: Financial burden

a) The problems related to the financial burden associated with your healthcare

Dimension 4: Lifestyle burden

a) The problems related to the burden related to dietary changes

b) The problems related to the burden related to doctors' recommendations to practice physical activity

Dimension 5: Social burden

a) How does your healthcare impact your relationship with others

b) "The need for medical healthcare on a regular basis reminds me of my health problems"

The original TBQ (referred to as TBQ-13 because of having 13 questions) was derived from a literature review and qualitative semi-structured interviews in France [36]. Later the questionnaire was validated in multiple English-speaking countries, including Canada, New Zealand, Australia, the UK, and the USA, which is referred to as TBQ-15 [36]. For this study, we used TBQ-15. The score for each question ranged from 0 to 10 whereby a score of 0 corresponds to no burden and 10 to a considerable burden. The global score of treatment burden ranges from 0 to 150; higher scores represent a higher level of burden. The global treatment burden score was considered as a continuous variable [36].

**Validity and reliability of TBQ.** During the development of the TBQ, the questionnaire was assessed for validity and reliability across multiple countries [36]. The authors of the TBQ found evidence supporting the construct validity for its use to assess the treatment burden for patients with at least one chronic condition in English-speaking countries. In addition to measuring construct validity, they established the evidence for the high reliability of the questionnaire by using the test-retest method at baseline and after 2 months. However, one limitation of this scale is that the validity of the scale was assessed among highly educated patient populations [12].

**Patient telehealth usage scale.** To measure patients' telehealth, use in the last 12 months, we used the Pew Research Center's telehealth use scale as a baseline scale from which items were selected and modified to form a new scale referred to here as the Patient Telehealth Usage Scale. The original Pew Research Center survey was created to measure the frequency of telehealth use among the US-based adult population [37]. It was the only telehealth use scale found through literature review and is used commonly in the patient-oriented literature [37].

Although there are many types of telehealth applications available for patients, for practical purposes, we have selected 3 types of telehealth services that patients with chronic conditions engage with frequently in Ontario-based primary care settings. These three services are recognized as telehealth by the Ontario Telehealth Network [38] for providing patient care: (1)

medical consultation or virtual visits; (2) Remote patient monitoring devices such as telehome care; and (3) Self-management using mHealth devices.

To report the frequency of different types of telehealth use, patients self-reported their frequency of telehealth use using the following scale: Never, once a month, Once a week, Several times a week, Once a day, Several times a day. This frequency scale was modified from Rosen et al's [39] Media and Technology Usage and Attitudes scale (MTUAS; S2 Appendix). We decided to modify the frequency level because patients with chronic disease, and especially patients with multimorbid conditions, may require monitoring their symptoms much more frequently than Rosen et al.'s scale suggests.

## Confounders

**Sociodemographic information.** To control for potential confounders, a range of sociodemographic characteristics including age, biological sex, income level, level of education, employment status, marital status, and presence of an unpaid caregiver (i.e., a person who is not an employed caregiver, such as a family member) was collected. Additionally, we collected the patient's number and types of chronic conditions. These variables were chosen because they may be associated with the patient's perceived treatment burden [14,29,30].

## Statistical analysis

Descriptive statistics, correlational analysis, t-tests and one-way ANOVAs, and hierarchical linear regression analysis were applied to analyze the survey data. The statistical analyses were performed using the R software version 4.1.1. Statistical significance was considered at $P < .05$ unless otherwise specified. All test results were 2-tailed.

A correlation analysis (Spearman Rho) was conducted to explore the association between three types of telehealth use (Communication, schedule, monitoring) and two categories of treatment burden (category 1: medication and lifestyle burden; category 2: administrative, financial and social burden). We categorized the medication and lifestyle burden together since these two types of burdens could contribute positively to patients' health. On the other hand, administrative, financial and social burdens do not directly contribute to patients' health, hence; these three dimensions of burdens collapsed together.

## Validity and Reliability of TBQ and telehealth usage scale for the target population

The validity of the newly created telehealth usage scales was established by the research team using the face validity method [40]. This involved evaluating the scales based on the research team's expertise in the subject matter, ensuring the relevance and appropriateness of questions. To test the reliability (internal consistency) of the TBQ and telehealth usage scale, we calculated Cronbach's alpha for both scales [40]. The overall reliability of the telehealth usage scale and TBQ was good, with a Cronbach's alpha of 0.85, and .80, respectively.

## Selecting variables for hierarchical regression analysis

We conducted a bivariate correlation analysis to detect which independent variables should be included in the subsequent regression analysis. As a large number of confounders may affect statistical power, we only included variables that were significantly correlated with the dependent or independent variables in the regression analysis described below. To avoid multicollinearity, if two independent variables were strongly correlated with each other, we only

included one of these variables in the regression analysis or created a composite score as described below.

We used one-way Analyses of Variance (ANOVA) and t-tests to identify differences in the frequency of telehealth use and treatment burden scores between demographic groups. Post-hoc analysis (Tukey's test) was used to assess pairwise comparisons if statistical significance was found between demographic groups in one-way ANOVA.

### Creation of composite score for SES index

To assess the SES of patients in our study, we constructed a composite SES measure, which integrated two SES-related components: income, and education [17]. This composite measure was developed because, income and education were highly correlated with each other, which can lead to multicollinearity. The creation of the SES composite measure involved 2 main steps:

**Assigning weights:** We assigned weights to each component to reflect their importance in determining SES. In this case, we assigned a weight of 0.6 to income and 0.4 to Education. The decision to weigh income higher than education was theory and data driven. Previous Canadian studies suggest that income is strongly correlated with patient health outcomes and social class [41,42], especially for older adults [17,43]. Since 61% of the patients in our study were retired, with an average age of 64.19, we anticipated that income may reflect their SES better than education.

**Calculating the SES index:** We then calculated the SES index for each patient by combining the scores from the income, and education variables. As both income level and the highest level of education are ordinal variables, this was done by multiplying each component score by its respective weight and summing these weighted scores.

### Composite measure of the frequency of telehealth use

Similarly, since there was a high correlation between three types of telehealth use: communicating, scheduling, and monitoring, a total frequency of telehealth use was created by combining the three ordinal variables [44]. Each participant received a total frequency of telehealth use score which was the sum of three types of telehealth use.

### Hierarchical linear regression analysis

To assess the association between treatment burden and frequency of telehealth use above and beyond demographic characteristics and socioeconomic variables, we used a hierarchical linear regression analysis [45] with three models: (1) model 1 predicted treatment burden score with age, sex, having a caregiver, and number of chronic conditions; (2) model 2 included two additional SES-related variables: SES index and employment status; (3) model 3 included one additional variable: the frequency of telehealth use. As is often the case, we treated ordinal variables (educational attainment, annual household income) as continuous in this analysis [46]. For categorical variables (sex, employment status, marital status, having a caregiver), dummy variables were created and used in the analysis. We reported adjusted beta coefficients, p-values, and 95% CI.

## Results

### Characteristics of study participants

A total of 75 patients completed the survey. The mean age of the participants was 64.19 (SD = 18.93), with 66.67% being female (n = 50). The sample represented a diverse range of

educational backgrounds and income levels. Eleven percent (n = 8) of patients had an annual household income of less than $25,000. Among the participants, 61% (n = 46) were retired, while 30.66% (n = 23) were employed. The average number of chronic conditions was 3.1 (SD = 1.03). The most prevalent chronic conditions included COPD (33%), diabetes (27%), chronic pain (22%), and hypertension (16%), underscoring the diverse health profiles within the sample. **Table 1** provides a detailed description of participant characteristics.

**Table 1. Descriptive demographic table.**

| Variables | N (%) |
|---|---:|
| **Age** (mean, SD) | 64.19 (18.93) |
| **Biological Sex** | |
| Female | 50 (66.67) |
| **Educational Attainment** | |
| Some high school, no diploma | 10 (13.33) |
| High school graduate, diploma or the equivalent | 10 (13.33) |
| Some college credit, no degree | 11 (14.66) |
| College degree | 12 (16.00) |
| Trade/technical/vocational training | 13 (17.33) |
| Graduate degree | 19 (25.33) |
| **Annual Household Income (CDN)** | |
| Less than 25,000 | 15 (20.00) |
| $25,000-$50,000 | 24 (32.00) |
| 50,000–100,000 | 26 (34.67) |
| 100,000–200,000 | 10 (13.33) |
| **SES Index (mean, SD)** | 3.58 (0.78) |
| **Employment status** | |
| Retired | 46 (61.33) |
| Employed | 23 (30.66) |
| Unemployed | 6 (8) |
| **Marital status** | |
| Single | 10 (13.33) |
| Married or domestic partnership | 44 (58.66) |
| Widowed | 10 (13.33) |
| Divorced | 11 (14.66) |
| **Have a caregiver** | 29 (38.67) |
| **Number of chronic conditions (mean, SD)** | 3.1 (1.03) |
| **Types of Chronic Conditions** | |
| COPD | 30 (40.00) |
| Diabetes | 24 (32.00) |
| Hypertension | 14 (18.67) |
| Stroke | 6 (8.00) |
| Cancer | 6 (8.00) |
| Depression | 14 (18.67) |
| Chronic pain | 20 (26.67) |
| Chronic Kidney Disease | 3 (4.00) |
| Arthritis | 20 (26.67) |
| Osteoporosis | 12 (16.00) |
| Other | 21 (28.00) |

**Table 2. Mean, median, and interquartile range of treatment burden questionnaire.**

| Variables | Mean (SD[b]) | Median (IQR[c]) |
|---|---|---|
| **Dimension 1. Medication Burden** | | |
| a) The problems related to the taste, shape, or size of your tablets and/or the annoyances caused by your injections | 3.51 (2.45) | 3 (4) |
| b) Number of times you should take your medication daily | 4.56 (2.88) | 5 (4) |
| c) The efforts you make not to forget to take your medications | 4.54 (2.76) | 5 (5) |
| d) The necessary precautions when taking your medications | 4.12 (2.29) | 5 (4) |
| **Dimension 2: Administrative Burden** | | |
| a) The problems related to lab tests and other exams (frequency, time spent, and associated nuisances or inconveniences | 5.02 (2.61) | 5 (3) |
| b) The problems related to self-monitoring: frequency, time spent, and associated nuisances or inconveniences | 4.12 (2.40) | 3.5 (3) |
| c) The problems related to doctor visits and other appointments: frequency and time spent for these visits and difficulties finding healthcare providers | 5.24 (2.38) | 5 (4) |
| d) How would you rate the problems related to the difficulties you could have in your relationships with healthcare providers (for example: feeling not listened to enough or not taken seriously) | 4.71 (2.47) | 5 (4) |
| e) The problems related to arranging medical appointments (doctors' visits, lab tests, and other exams) and reorganizing your schedule around these appointments | 4.98 (2.75) | 5 (5) |
| f) The problems related to the administrative burden related to healthcare (for example: all you have to do for hospitalizations, reimbursements, and/or obtaining social services) | 5.43 (2.77) | 6 (5) |
| **Dimension 3: Financial burden** | | |
| a) The problems related to the financial burden associated with your healthcare | 5.62 (2.95) | 7 (5.25) |
| **Dimension 4: Lifestyle burden** | | |
| a) The problems related to the burden related to dietary changes | 4.39 (2.68) | 5 (4) |
| b) The problems related to the burden related to doctors' recommendations to practice physical activity | 4.97 (2.61) | 5 (4.75) |
| **Dimension 5: Social burden** | | |
| a) How does your healthcare impact your relationships with others? | 5.45 (2.79) | 5 (4.75) |
| b) "The need for medical healthcare on a regular basis reminds me of my health problems" | 5.85 (2.39) | 6 (5) |

*All items were measured from 0 to 10

[b]SD = Standard deviation

[c]IQR = Interquartile Range

## Description of patient experienced treatment burden

The average treatment burden score was: 72.15 (SD: 26.18) out of a maximum of 150. **Table 2** shows the mean, standard deviation, median, and interquartile range of each dimension of the scale. Each dimension was rated on a scale of 0 to 10.

## Profile on Telehealth use in the last 12 months

Among the 75 participants, a significant majority, 85% (64 out of 75), reported utilizing telehealth for communication with their primary care team over the past 12 months. Similarly, 72% (54 out of 75) used telehealth for scheduling or rescheduling medical appointments. However, a lower percentage, 35% (26 out of 75), reported using telehealth for self-management purposes.

The frequency of telehealth for communication with the primary care team varied, with 37% reporting never using it, while 39% used it once a month, and 9% utilized it several times

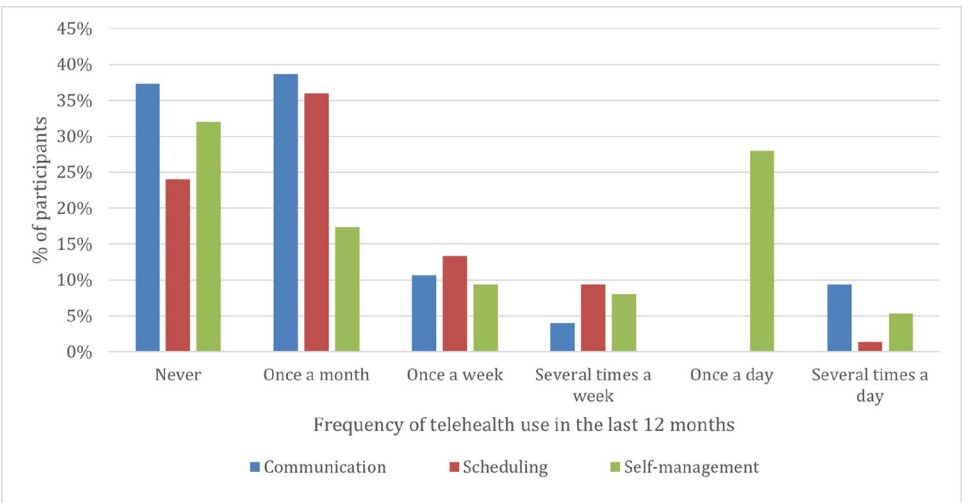

**Fig 2. Profile of telehealth use for communication, scheduling, and self-management purposes.**

a day. Similarly, the frequency of telehealth for scheduling or rescheduling medical appointments varied, with 24% reporting never using it and 36% using it once a month. For self-management, 32% reported never using it, while 17% used it once a month, and 28% utilized it at least once a day. **Fig 2** shows a detailed description of telehealth use.

**Correlation analysis.** Correlation analysis (**Table 3**) was used to explore the association between demographic factors (Age, number of chronic conditions, total frequency of telehealth use) and treatment burden. The analysis demonstrated that age (r = -.54; p < .001), number of chronic conditions (r = .37; p < .001), and total frequency of telehealth (r = .40; p < .001), were significantly correlated with treatment burden. However, there was no statistical significance of the SES index with other variables.

The correlation between types of telehealth use and two types of treatment burden (medication and lifestyle; administrative, financial, and social) is presented in Table 4. The analysis demonstrated that the frequency of communication (ρ = .38; p < .001), frequency of scheduling (ρ = .31; p < .001), and frequency of monitoring (ρ = .35; p < .001), were significantly correlated with the medication and lifestyle burden. However, there was no statistical significance of the administrative, financial and social burden with any category of telehealth use.

Additionally, we conducted ANOVA and t-test to identify differences in either total frequency of technology use or treatment burden score between groups of demographic variables:

**Table 3. Correlation (Pearson r) between the numeric variables.**

| Variables | 1 | 2 | 3 | 4 | 5 |
|---|---|---|---|---|---|
| 1. Age | _ | | | | |
| 2. Number of chronic conditions | -.10 | _ | | | |
| 3. Frequency of telehealth use | -.21 | .32* | _ | | |
| 4. Treatment burden | -.54* | .37* | .40* | _ | |
| 5. SES index | -0.18 | -0.15 | -0.09 | -0.14 | _ |

*Indicates statistical significance

**Table 4. Correlation analysis (Spearman rho) between types of technology use and two categories treatment burden.**

| Types of burden | Frequency of communication | Frequency of Scheduling | Frequency of Monitoring |
|---|---|---|---|
| Medication+Lifestyle burden | 0.38* | 0.38* | 0.38* |
| Administrative+Financial+Social Burden | 0.18 | 0.18 | 0.18 |

*Indicates statistical significance

A one-way ANOVA indicated statistically significant differences in treatment burden scores based on participant's employment status (F (2, 69) = 10.03, p < .01). Further post-hoc tests indicated significant differences in treatment burden scores between employed and retired individuals (mean difference = 27.23, p < .01). However, it did not identify a significant difference between unemployed-retired individuals or unemployed-employed individuals.

Additionally, we did not find a statistically significant difference in treatment burden based on marital status (one-way ANOVA). Hence, we did not add marital status to the regression model.

There was a significant difference between those with unpaid caregivers and those without t (72) = 6.91, p < .01 (t-test). Hence, the presence of the caregiver was added to the regression model.

## Association between participants' demography, frequency of telehealth use, and treatment burden

We used a hierarchical linear regression analysis, to further assess the association between tele-health use and treatment burden above and beyond demographic and SES-related variables.

In model 1, demographic variables (excluding SES-related variables) were significant predictors of treatment burden (Table 5, model 1), and accounted for 51% of the variance in treatment burden. Similarly, SES-related variables (model 2) and the frequency of telehealth use (model 3) accounted for an additional 3% each of the explained variance in treatment burden score.

**Table 5. Hierarchical linear regression predicting the treatment burden.**

| Variables | Model 1*** | Model 2*** | Model 3*** |
|---|---|---|---|
| **Age** | -.40* | -0.47* | -0.48* |
| **Male Sex** | -17.05*** | -12.44* | -11.81 |
| **Have a caregiver** | 16.34* | 20.24*** | 19.12*** |
| **Number of Chronic Conditions** | 5.22*** | 2.55 | 1.48 |
| **SES Index** | | -6.16* | -5.87 |
| **Employment** (Ref: Unemployed) | | | |
| Employed | | -8.97 | -5.49 |
| Retired | | -10.93 | -5.82 |
| **Total frequency of telehealth use** | | | 0.99 |
| **R^2** | .51 | .54 | .57 |
| **R^2 Change** | .51 | 0.03 | 0.03 |
| **Delta F** | 18.01 | 11.59 | 10.79 |

Note

*p<0.1

**p<0.05

***p<0.01

The frequency of telehealth use was positively associated with treatment burden but was not statistically significant after controlling for demographic and SES-related variables (Table 5, model 3). Overall, the combination of demographic variables, SES-related variables, and frequency of total technology use explained 57% of the variance in treatment burden score (Table 5, Model 3). Having a caregiver (positively) and age (negatively) were independently associated with a high treatment burden, even after controlling for other variables (Table 5, model 3).

## Discussion

The primary objective of this study was to explore whether and to what extent telehealth use is associated with treatment burden. The secondary objective of this study was to identify demographic and SES-related variables that are associated with treatment burden.

Our hypothesis (H1) anticipated a negative relationship between telehealth utilization and treatment burden, implying that higher use of telehealth would be linked to reduced treatment burden [29]. This hypothesis was created based on prior literature that suggests telehealth can reduce time, travel, and cost-related burdens. Contrary to this hypothesis, in our analysis there was a positive significant correlation between the frequency of telehealth use and treatment burden scores. Even after adjusting for demographic and socio-economic factors in the hierarchical regression, the positive association persisted; however, it was not statistically significant. This reduced significance might be attributed to the inclusion of demographic and socio-economic variables, potentially reducing the statistical power of the model, especially given our small sample size.

Further analysis shows that higher use of telehealth use is positively associated with both 'health positive' burden (medication, and lifestyle) and non-beneficial burden (administrative, financial travel). However, the association between high telehealth use and 'health positive' burden was statistically significant, as opposed to the non-beneficial burden. This finding partially supports our hypothesis (H1a) which anticipated that telehealth use will be positively associated with treatment burden that is beneficial for health (i.e., following medication regimen and lifestyle) but negatively associated with administrative, financial and travel burden. Overall, the significant association between telehealth use and 'health positive' burdens likely reflects the primary healthcare goals of telehealth—improving patient outcomes through better management and adherence to treatment plans such as medication adherence or adherence to exercise or diet regimen. Conversely, the non-significant association with non-beneficial burdens suggests that telehealth's impact on these aspects is less direct or has not been achieved yet.

Despite the lack of statistical significance, the positive link between high telehealth utilization and treatment burden that was identified in our analysis warrants further exploration in future studies. Existing literature presents a nuanced perspective on the relationship between telehealth use and treatment burden. Notably, a previous Canadian study suggested that virtual care was associated with reduced treatment burden in patients with chronic conditions [29]. On the other hand, a comprehensive systematic review of cancer survivors indicated mixed results, emphasizing the impact of required telehealth frequency. This study posits that frequent telehealth engagement, especially on a weekly or daily basis, may increase the treatment burden [23]. Another umbrella review confirmed patients' concerns about the time burden associated with digital health interventions (including telehealth) [47]. In sum, multiple studies suggest that patients' time scarcity is a major concern when it comes to integrating digital health into their lives.

This time scarcity can be a plausible explanation for the positive association between the frequency of telehealth use and the treatment burden in our patient population. It is especially important for patients with low SES who may have lower available discretionary time and lower agency over time due to routine fragmentation (e.g. multiple part-time jobs, inflexible work schedules), higher caregiving responsibilities (e.g. lack of access to paid caregiver) [48] and neighbourhood disorganization (e.g. community violence) [49,50]. Furthermore, at the macro-level, this patient population needs to spend extra time to mitigate the effects of deprivation caused by living in neighbourhoods with low public investment (e.g. lack of public transportation, lack of jobs, health centers) [50]. Given that 52% of our study participants had annual household income of less than $50,000 CDN along with 28% of participants using self-management devices at least once a day, this patient population may face additional time constraints related to self-management, and ultimately leading to higher treatment burden. However, it is important to note that the cross-sectional nature of our study restricts us from drawing definitive conclusions about this association.

Our second hypothesis anticipated that both telehealth use and treatment burden will be influenced by patient sociodemographic characteristics [14,29,30]. This hypothesis was partially supported in our analysis. Model 3 of the regression analysis shows that having an unpaid caregiver and younger age were associated with a higher treatment burden.

The association between unpaid caregivers and high treatment burden has been confirmed by a previous cross-sectional study [11]. It is plausible that this patient population has substantial care needs (e.g. complex multimorbidity, frailty) [51], which leads to a high treatment burden necessitating support from family or friends [52]. However, due to the cross-sectional nature of this paper, we cannot establish causality between these two variables.

Additionally, our findings show that younger age is associated with a higher treatment burden. This is consistent with the findings of a previous large-scale Australian cross-sectional study [11]. This association between younger age and high treatment burden may be attributed to the higher likelihood of younger individuals being employed, an explanation that is partially supported by our ANOVA, and having work and family responsibilities such as childcare [53]. These multiple responsibilities might make having chronic conditions more disruptive to the lifestyle of younger people [54]. Alternatively, it could imply that older patients become accustomed to their care-related tasks over time, finding them less burdensome, or that they have more time to manage them upon retirement [55]. Exploring the reasons behind the association of younger age with an increased burden warrants further investigation.

Moreover, in models 1 and 2 we see that the female sex is associated with a high treatment burden, which is confirmed in the previous literature [11,12]. Female patients often navigate dual roles as caregivers and chronic patients, coupled with additional daily life burdens like domestic chores, potentially explaining this phenomenon [56,57].

Overall, telehealth can significantly reduce the administrative and financial burdens for patients with chronic conditions. Patients with low SES, in particular, can benefit from the efficiencies offered by telehealth, such as reduced paperwork and travel. However, they may also experience better health outcomes due to higher engagement with health-promoting activities. Ensuring equitable access and making the technology easy to use for these specific populations is crucial to realizing these benefits.

## Limitations and future directions

Our study has a few limitations, including its cross-sectional nature, which limits our ability to establish causation. While our study identifies associations and patterns within the collected data, it cannot definitively ascertain the direction of causality or temporal sequencing of

outcomes. Moreover, the limited sample size might have prevented the identification of additional associations, particularly those with small effect sizes. The small sample size may also affect the generalizability of the findings. Moreover, the recruitment material and data collection tools were only available in English, hence, Francophone Ontarians and immigrants may be underrepresented in the current study. Future research should explore these relationships in larger and more diverse samples and employ longitudinal designs to better understand the dynamic nature of treatment burden over time.

Additionally, although we specifically recruited participants through CHCs to target patients with low SES, 13% of our study population had incomes over 100,000 CAD. Previous studies have highlighted the difficulty of recruiting low SES patients for research. Moreover, we cannot differentiate the characteristics of those recruited from CHCs versus social media, as the surveys were anonymous, and this data was not collected. Additionally, although we offered a paper-based survey option to capture the experiences of patients who may not be technologically savvy, all participants chose the online survey version. This may limit the generalizability of the study findings to less tech-savvy patients. Furthermore, our study did not collect information such as the specific technology used by patients as well as patients' attitudes towards technology use. Both of these factors could potentially impact patients' experienced treatment burden while using telehealth.

Finally, the TBQ-15 scale has a limitation in its distribution of questions, with a notable emphasis on medication and administrative burdens, as opposed to the relatively fewer questions addressing financial, lifestyle, and social burdens [12]. This imbalance may impact the interpretability of our study results.

## Conclusion

In conclusion, this study highlights multiple demographic factors that are associated with treatment burden. It suggests a positive relationship between telehealth use and treatment burden. Further investigation is needed to have a clear understanding of how to make telehealth interventions more patient-oriented and less burdensome for patients and their caregivers.

## Supporting information

**S1 Appendix. Cross-sectional Study-Treatment burden questionnaire.**
(PDF)

**S2 Appendix. Telehealth usage scale.**
(PDF)

## Author Contributions

**Conceptualization:** Farah Tahsin, Carolyn Steele Gray, Jay Shaw, Aviv Shachak.

**Formal analysis:** Farah Tahsin.

**Supervision:** Carolyn Steele Gray, Aviv Shachak.

**Visualization:** Farah Tahsin.

**Writing – original draft:** Farah Tahsin, Carolyn Steele Gray, Aviv Shachak.

**Writing – review & editing:** Farah Tahsin, Carolyn Steele Gray, Jay Shaw, Aviv Shachak.

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
