## [Decision Letter · Decision Letter 0]

10 Jun 2024

PDIG-D-24-00138

Exploring the Relationship Between Telehealth Utilization and Treatment Burden among Patients with Chronic Conditions: A Cross-Sectional Study in Ontario, Canada

PLOS Digital Health

Dear Dr. Tahsin,

Thank you for submitting your manuscript to PLOS Digital Health. After careful consideration, we feel that it has merit but does not fully meet PLOS Digital Health's publication criteria as it currently stands. Therefore, we invite you to submit a revised version of the manuscript that addresses the points raised during the review process.

Please submit your revised manuscript within 60 days Aug 09 2024 11:59PM. If you will need more time than this to complete your revisions, please reply to this message or contact the journal office at digitalhealth@plos.org. Please include the following items when submitting your revised manuscript:

We look forward to receiving your revised manuscript.

Kind regards,

Calvin Or, PhD

Section Editor

PLOS Digital Health

Journal Requirements:

Additional Editor Comments (if provided):

Reviewers' comments:

Reviewer's Responses to Questions

**Comments to the Author**

1. Does this manuscript meet PLOS Digital Health’s publication criteria? Is the manuscript technically sound, and do the data support the conclusions? The manuscript must describe methodologically and ethically rigorous research with conclusions that are appropriately drawn based on the data presented.

Reviewer #1: Partly

Reviewer #2: Yes

2. Has the statistical analysis been performed appropriately and rigorously?

Reviewer #1: Yes

Reviewer #2: Yes

3. Have the authors made all data underlying the findings in their manuscript fully available (please refer to the Data Availability Statement at the start of the manuscript PDF file)?

Reviewer #1: Yes

Reviewer #2: No

4. Is the manuscript presented in an intelligible fashion and written in standard English?

Reviewer #1: Yes

Reviewer #2: Yes

5. Review Comments to the Author

Reviewer #1: The paper authors asks a very relevant and timely question about whether the increased use of telehealth offloads the treatment burden onto patients with chronic diseases, from healthcare providers. They do it by evaluating the association between telehealth use and burden a survey (Treatment Burden Questionnaire (TBQ-15)) to patients with chronic diseases. They used a hierarchical linear regression analysis, to further assess the association between telehealth use and treatment burden above and beyond demographic and SES-related variables. They also look at the effects of SES on this burden association. The frequency of telehealth use had a positive direction association with treatment burden but was not statistically significant after controlling for demographic and SES-related variables.

The key issue to me is the definition or inclusion of factors for ‘treatment burden’ may be simultaneously both positive (e.g. more exercise is included as a ‘burden’ or taking their medications which if done by the patient is better for diabetes) and negative (e.g. more time trying to navigate technology may be excessive or detrimental to the patient), so it is difficult to make a conclusion without teasing out these aspects. It would be more interpretable if the analysis was done by either positive or negative to the patient’s health condition, or factors that must be the patient’s responsibility vs factors that could be absorbed by the provider side instead of the patient. (e.g. efficiency and effectiveness both considered).

If these aspects were teased out the paper would be very interesting and an informative addition to the current literature. 

Comments by section:

Abstract:

Mean or median plus SD or range should be shown for age and 2 decimal point seems excessive.

The results should show the key findings (associations) in numbers as well as narrative with appropriate statistical parameters.

Introduction:

Please delineate positive versus negative factors (e.g. exercise vs trouble shooting technology) and those that are require patient involvement for their health (e.g. taking medications or exercise vs using technology or attending appointments) vs that contribute to a patients treatment burden. For instance, if telehealth causes the patient to exercise more hours at home but have to spend less time taking the bus to clinic, is that an increase in burden?

Similarly, are the SES issues related to access or burden? Differentiate these. If a lower SES patient is less often offered video telehealth because they don’t have a phone or internet, is the telehealth a burden to them if they don’t use it? Limited “access” is not the same as “burden”. 

Would it be considered mitigating the burden to the patient of requiring exercise three times a week supported by telehealth if the patient simply chose not to exercise? This differential value needs to be considered. It would be less burden but less desirable/effective. 

Methods:

Study population – please indicate if the CHC and social media recruited patients were similar or different given comments on CHS marginalized focus. Also, 13% of patients made >$100k. Describe.

If patients were offered an online survey option, did this bias the recruitment to more technology capable participants? Similarly, how did the methods try to avoid recruitment bias for participation capable, available, or willing to participate in research (e.g. were there other research processes that bias for or against certain SES?)

In the TBQ-15, Dimension 1 would be expected to increase if the patient is adherent to medications, which is desirable. Similar with 4. The other Dimensions, especially Administrative and Financial burden decreasing the burden would be desirable by telehealth. Some, e.g. 5 could go either way. 

Explain why mHealth self-management applications are included together with HCP virtual care applications as they may have different usage and burden expectations. For example, virtual consults from a healthcare provider would be expected to potentially reduce patient travel and other time, but using an mHealth self-management tool would be expected to increase the patient burden, even if it is in a “good” (beneficial requirement) burden time such as helping them take their medications properly or exercising which is their health goal. Similarly for frequency – HCP virtual care tools would be expected to be only required episodically for visits/consultations, whereas a patient self-management app might be expected to be used daily (in a positive way). These should be differentiated and discussed.

The methods would be improved if they evaluated the 5 dimensions of the TBQ-15 as independent or logically grouped associations and if they separated self-management tools in sensitivity analysis. Although, this would likely require a larger sample size to be able to detect significant associations and differences. (already under an ideal sample size to detect significant diferences). 

Results:

It is unclear what the individual burden scores reflect as only the total score was described out of 150: “On average, participants reported the highest burden in their daily lives due to regular healthcare needs (mean=5.85), followed by financial aspects (mean=5.62), social relationships (mean=5.45), and administrative work (mean=5.43).” Please describe (and the differences seem tiny)

Table 2 is very helpful to see the types of use, but as above, they have different ‘burden’ purposes.

Discussion:

As the authors acknowledge, their “hypothesis (H1) anticipated a negative relationship between telehealth utilization and treatment burden, implying that higher use of telehealth would be linked to reduced treatment burden.” And that “This hypothesis was created based on prior literature that suggests telehealth can reduce time, travel, and cost-related burdens.” 

Therefore only burdens related to time, travel, and cost would be expected to be lower burden, but if the patient adheres to their treatment recommendations, we would expect medication and other burdens to increase. It is important to know which factors are affected in which direction.

I don’t think the goals of telehealth is to have patients spend less time (burden) on self-management, rather only on the aspects of care that are not required to improve their health, such as some areas of logistics.

In fact, contrary to the authors interpretation, I would posit that people of lower SES would stand to benefit even greater from certain efficiencies offered by various forms of telehealth, at the same time also benefit from greater ‘burden’ (time and effort) of engagement in self-management of their chronic conditions so they reach their health goals. However, equity of access and appropriateness of ease of use of the technology for the specific populations would be key to achieve those goals. 

This perspective should be discussed to balance their argument.

Reviewer #2: One very important piece that this article misses is records of the type of phones people used to access telehealth services. This can affect user experience and ultimately impact on treatment burden. Secondly, a question about the participants use behavior of smartphones and computers can impact on this as well. Not everyone like computers or phones regardless of their SES.

6. PLOS authors have the option to publish the peer review history of their article (what does this mean?). If published, this will include your full peer review and any attached files.

**Do you want your identity to be public for this peer review?** For information about this choice, including consent withdrawal, please see our Privacy Policy.

Reviewer #1: Yes: Richard Lester

Reviewer #2: Yes: Kingsley I. Ndoh, MD, MPH

---

## [Decision Letter · Decision Letter 1]

7 Aug 2024

Exploring the Relationship Between Telehealth Utilization and Treatment Burden among Patients with Chronic Conditions: A Cross-Sectional Study in Ontario, Canada

PDIG-D-24-00138R1

Dear Ms Tahsin,

We are pleased to inform you that your manuscript 'Exploring the Relationship Between Telehealth Utilization and Treatment Burden among Patients with Chronic Conditions: A Cross-Sectional Study in Ontario, Canada' has been provisionally accepted for publication in PLOS Digital Health.

Best regards,

Calvin Or, PhD

Section Editor

PLOS Digital Health

Reviewer Comments (if any, and for reference):

Reviewer's Responses to Questions

**Comments to the Author**

1. If the authors have adequately addressed your comments raised in a previous round of review and you feel that this manuscript is now acceptable for publication, you may indicate that here to bypass the “Comments to the Author” section, enter your conflict of interest statement in the “Confidential to Editor” section, and submit your "Accept" recommendation.

Reviewer #1: All comments have been addressed

2. Does this manuscript meet PLOS Digital Health’s publication criteria? Is the manuscript technically sound, and do the data support the conclusions? The manuscript must describe methodologically and ethically rigorous research with conclusions that are appropriately drawn based on the data presented.

Reviewer #1: Yes

3. Has the statistical analysis been performed appropriately and rigorously?

Reviewer #1: Yes

4. Have the authors made all data underlying the findings in their manuscript fully available (please refer to the Data Availability Statement at the start of the manuscript PDF file)?

Reviewer #1: Yes

5. Is the manuscript presented in an intelligible fashion and written in standard English?

Reviewer #1: Yes

6. Review Comments to the Author

Reviewer #1: The authors have sufficiently addressed my key concerns and improved the paper.

7. PLOS authors have the option to publish the peer review history of their article (what does this mean?). If published, this will include your full peer review and any attached files.

**Do you want your identity to be public for this peer review?** For information about this choice, including consent withdrawal, please see our Privacy Policy.

Reviewer #1: **Yes: **Dr. Richard T Lester, MD, FRCPC
